# Synergistic Effect of SiO_2_ and Fe_3_O_4_ Nanoparticles in Autophagy Modulation

**DOI:** 10.3390/nano14121033

**Published:** 2024-06-15

**Authors:** Sitansu Sekhar Nanda, Danyeong Kim, Hyewon Yang, Seong Soo A. An, Dong Kee Yi

**Affiliations:** 1Department of Chemistry, Myongji University, Yongin 17058, Republic of Korea; nandasitansusekhar@gmail.com; 2Department of Bionanotechnology, Gachon Medical Research Institute, Gachon University, Seongnam 13120, Republic of Korea; dan627328@gmail.com (D.K.); yhw0528@gachon.ac.kr (H.Y.)

**Keywords:** silica, iron oxide, synergistic effect, ROS, autophagy

## Abstract

Rapid advancements in nanotechnology have expanded its applications and synergistic impact on modern nanosystems. The comprehensive assessment of nanomaterials’ safety for human exposure has become crucial and heightened. In addition to the characterization of cell proliferation and apoptosis, probing the implication of autophagy is vital for understanding the ramification of nanomaterials. Hence, HEK-293 kidney cells were employed to understand the changes in induction and perturbation of autophagy in cells by iron oxide (Fe_3_O_4_) and silica (SiO_2_) nanoparticles. Interestingly, Fe_3_O_4_ worked as a potent modulator of the autophagy process through its catalytic performance, which can develop better than that of SiO_2_ nanoparticles mechanism, stressing their therapeutic implication in the understanding of cell behaviors. The quantification of reactive oxygen species (ROS) was measured along with the process of autophagy during cell growth. This modulated autophagy will help in cell fate determination in complementary therapy for disease treatment, provide a clinical strategy for future study.

## 1. Introduction

Nanomaterials exhibit unique optical and physico-chemical properties, such as fluorescent brightness of quantum dots [1], localized surface plasmon resonance (LSPR) of gold nanorods [2], and superparamagnetic effect of iron oxide nanoparticles [3]. For advanced theranostics, the influence of metallic nanoparticles (NPs) on the differentiation and proliferation of epidermal, cancer, and normal cells was investigated in depth [4,5,6,7,8]. For the safe development and commercial use of NPs, phototoxicity and cellular toxicity investigations are essential requirements. Previously, NP mediated toxicity was observed in diverse physiological areas, such as the inhibition of cell division, cell death, genetic damage, inflammation and oxidative stress [9,10]. Different mechanisms could occur in metallic nanoparticle-induced cellular differentiation and proliferation, such as regulation/interaction of specific transcription factors, providing materials for reactive oxygen species, and modulation of the intercellular process [11]. Inflammatory cell expression, such as that of macrophages and neutrophils, was associated with metallic nanoparticle related toxicity, because the generation of ROS was closely influenced by the inflammatory cell expression [12]. This was also considered as free radical facilitated toxicity by the Fenton reaction [13]. A detailed study of the application of metallic NPs to cellular differentiation and proliferation could lead to better design and preparation of metallic NPs for future scientific applications in regenerative medicine and for the modulation of cell functions [14].

The SiO_2_ surface is one of the most important biocompatible interfaces that could be implemented in analyses of protein corona [15], bioimaging [16], drug delivery and diagnosis, due to its excellent physico-chemical stability, tunable pore structure and high surface area [17]. Orthosilicic acid, a soluble state of SiO_2_, does not alter the preparation of alkaline phosphate for cartilage production. Another soluble state of SiO_2_ was also used in the preparation of alkaline phosphate and collagen [18]. SiO_2_ based nanomaterials were relevant and biocompatible for cell proliferation [19]. SiO_2_ could prevent bacterial infection of stem cell during cell growth [20], and SiO_2_ also promoted the osteogenic differentiation and proliferation of kidney cells [21]. SiO_2_ could induce the autophagy process, such as inducing autophagy dysfunction in L-02 and HepG2 cells [22,23], enhancing autophagic activity [24] and related phenomena [25,26] without a coherent mechanism. Fe_3_O_4_ with paramagnetic function is attractive for various biomedical applications and medical technologies [27]. Fe_3_O_4_ NPs obeyed Coulomb’s law and helped in transportation or immobilization of magnetically designed biological molecules. It initiated a biochemical reaction or modulation by attachment to mechanosensitive channels of the cell membrane. Fe_3_O_4_ could also be applied in autophagy related experiments, acting as an incidental factor [28,29]. The toxicity of Fe_3_O_4_, its abnormal cellular ROS balance and structural injury to mitochondria seemed to be interrelated [29]. Mitochondrial injury contributed to stimulation of autophagy.

In the biomedical fields, silica and iron oxide are the most common nanomaterials for drug delivery, tissue engineering, and cell promotion [30]. After entering cells, the NPs would degrade into irons and induce reactive oxygen species (ROS) through the changing of mitochondrial related organelles’ structures [31]. The surplus production of ROS could create a chain reaction inside, along with the autophagy process of the cell, which would consist of a process of discarding superfluous and unnecessary materials from cells and regulating the ROS homeostasis [32]. Autophagy plays a significant role in controlling cell behaviors. Excessive or inadequate levels of autophagy inside cells would boost cell aging, inhibit cell growth, and encourage cell apoptosis. However, the physiological balance of autophagy could reduce cell aging and improve cell growth and proliferation [33]. The advantage of combined SiO_2_ and Fe_3_O_4_ NPs could advance experimental efficiency and cut production costs. In the cellular process, combined SiO_2_ and Fe_3_O_4_ NPs could promote autophagy activity, an essential cellular degradation access. Recent studies revealed that combined NPs could indeed induce autophagy in cells [34,35]. Autophagy always plays a crucial role in the cell’s fate, favoring either death or survival [36,37,38], by disassembling the dysfunctional or unnecessary components [39,40]. Since nanomaterials could activate autophagy, the assessment of organelles and cellular proteins delivered to lysosomes would determine the fate of digestion by lysosomal hydrolases [41,42,43,44].

Anghelache et. al. [45] demonstrated that dextran-coated iron oxide nanoparticles have anti-inflammatory activity at non-cytotoxic concentrations by reducing the levels of pro-inflammatory mediators, such as IL-1β, MCP-1, CCR2, TNF-α, and IL-6, in activated endothelial cells and M1 phenotype macrophages. Folic acid-coated iron oxide nanoparticles enhance internalization and facilitate delivery of therapeutic agents for cancer treatment and inflammation-related diseases, like rheumatoid arthritis, lupus, osteoarthritis, Crohn’s disease, and atherosclerosis [46]. In this study, the possibility of assessing the synergistic effects of Fe_3_O_4_ and SiO_2_ NPs were explored with HEK-293 kidney cells. Fe_3_O_4_ NPs could offer superior biocompatible catalytic performance due to the enhancement provided by SiO_2_ NPs.

## 2. Materials and Methods

### 2.1. Materials

Igepal, cyclohexane, ammonium hydroxide (NH_4_OH, 28 wt% in water), tetraethyl orthosilicate (TEOS), iron (II) chlorides (FeCl_2_), iron (III) chlorides (FeCl_3_) and 3,4-diaminobenzoic acid (DABA) were purchased from Sigma-Aldrich, Saint louis, MO, USA. HEK-293 kidney cells were supplied from Gachon University.

### 2.2. Synthesis of SiO_2_

Dissolve 0.225 g Igepal in 4.4 mL cyclohexane. Then, 0.046 mL DI water was added to the mixed solution, stirring for 20 min. Then, 0.05 mL NH_4_OH was added, stirring for 20 min. Finally, 100 uL TEOS was added, stirring for 24 h [47]. 

### 2.3. Synthesis of Fe_3_O_4_

To prepare single-surfactant-coated Fe_3_O_4_, 10 mg iron (II), 20 mg (III) chlorides and 90 mg of DABA were added to 10 mL distilled water with magnetic stirring for 80 min at 80 °C. Next, 1 mL ammonium hydroxide was added to the mixture with magnetic bar stirring for 80 min at 85 °C. After the mixture was cooled to room temperature, iron oxide NPs were collected via magnetic bar and washed with distilled water and ethanol 3 times, respectively [48]. 

### 2.4. Characterization of SiO_2_ and Fe_3_O_4_

TEM (Vanila Inc., Salt Lake City, UT, USA, JSM5700F) and zetasizer (Cananda CARRY 40, Cananda) were used to characterize the shape and size of silica and iron oxide NPs.

### 2.5. Generation of LC3 Cell Line

Human embryonic kidney 293 cells (HEK-293; ATCC, CRL-1573) were sustained in Dulbecco’s modified eagle media (DMEM; Gibco™, Thermo Fisher Scientific, Waltham, MA, USA, 10566016) containing 10% fetal bovine serum (FBS; Gibco™, 16000044), 1% kanamycin Sulfate (Gibco™, 15160054), and 1% penicillin streptomycin (Gibco™, 15140122), and kept at 37 °C and 5% CO_2_. 

LC3 transfection was conducted using GFP–LC3 Expression Vector (CELL BIOLABS, San Diego, CA, USA, CBA-401). Briefly, experiments were performed after growing at 80–90% cell confluence. Cells were detached by Trypsin-EDTA (0.25%) (Gibco™, 25200072) and centrifuged at 1200 rpm for 3 min. The supernatant was discarded and resuspended in new media. The cells were collected (0.5 × 10^6^ cells) and centrifuged (300 g for 5 min). The supernatant was discarded and washed with DPBS (WELGENE, Busan, Republic of Korea, LB 001-02). After centrifuge, cells were resuspended with DPBS. Then, 1 µg GFP-LC3 DNA was added to the cells. Transfection was performed with the Neon™ Transfection System, which is an electroporation system (Invitrogen™, Carlsbad, CA, USA). After transfection, the cells were seeded to a 6-well plate without antibiotic. Selection cells were proceeded with Geneticin™ Selective Antibiotic (G418 Sulfate; Gibco™, 10131035). G418 was treated at 800 µg/mL for 3 days and changed to a decreased concentration (200 µg/mL). 

Success of transfection was confirmed by Real-time polymerase chain reaction (RT–qPCR). The cells were seeded to the 6-well plate and grown for 48 h. Then, RNA was extracted using an Accuprep universal RNA extraction kit (Bioneer, Daejeon, Republic of Korea, K-3141). Following extraction, RNA was converted to cDNA utilizing the AccuPower^®^ RocketScript™ Cycle RT PreMix (dT20) (Bioneer, K-2201). Subsequently, RT–PCR was conducted, employing the SYBR green system with AccuPower^®^ 2X GreenStar™ qPCR Master Mix (Bioneer, K-6251). Primer sequences for RT–qPCR, procured from Bioneer (Daejeon, Republic of Korea), are as follows: Forward: 5′-GAGCAGCATCCAACCAAA-3′; Reverse: 5′-CGTCTCCTGGAGGCATA-3′.

### 2.6. Cell Viability of NPs

LC3 cells were seeded at 1 × 10^4^ cells/well in a 96-well plate and stabilized for 24 h. The medium was discarded and NPs added to this mixture according to concentrations (1, 10, 20, 50, 100 µg/mL) in DMEM without FBS. After incubation for 48 h, NPs were removed, and cell viability reagent with DMEM (BIOMAX, Gyeonggi-do, Republic of Korea, QM10000) was added. The solution was treated for 1 h and optical density measured at 450 nm wavelength using a VICTOR 3™ multi-spectrophotometer (PerkinElmer, Waltham, MA, USA).

### 2.7. Reactive Oxygen Species (ROS) Study

The production of ROS was checked by the oxidant-sensitive dye 2′,7′-Dichlorofluorescin Diacetate (DCF–DA; Sigma-Aldrich, St. Louis, MO, USA, 287810). LC3–GFP transfected cells were seeded at 2 × 10^4^ cells/well in 96-well plates and incubated for 24 h. Cells were then treated with 100 µg/mL Fe_3_O_4_, SiO_2_, and these mixtures incubated for 24 h. As positive control, 400 μM H_2_O_2_ was treated in cells for 1 h. 

After incubation, the media was discarded carefully and 20 μM DCF-DA was treated for 30 min. Cells were washed with PBS and fluorescence intensity was measured using a VICTOR 3™ multi-spectrophotometer (PerkinElmer, Waltham, MA, USA) at the excitation and emission wavelengths of 485 nm and 535 nm.

### 2.8. Difference of LC3 Expression on NPs

Western blot was carried out for confirmation of LC3 protein. The cells were seeded at 1 × 10^6^ cells/well on a 6-well plate and stabilized for 24 h. Then, 50 µg/mL Fe_3_O_4_, SiO_2_ and these mixtures were added to the well and incubated for another 24 h. Cells were detached using Trypsin-EDTA (0.25%) and centrifuged at 1200 rpm for 3 min. Extraction of lysate proceeded using M-PER™ Mammalian Protein Extraction Reagent (Thermo Scientific™, Waltham, MA, USA, 78503) following the manufacturer’s instruction. The protein concentration was measured by Pierce™ BCA Protein Assay Kits (Thermo Scientific™, 23225). 

100 µg/mL protein was mixed with 4× Laemmli Sample Buffer (Bio-Rad, Hercules, CA, USA, 161-0747) and heated at 98 °C for 5 min. The samples were loaded in 20 µL/wells in 12% Tris/glycine gel and run at 100 V for 1 h after 50 V for 5 min. The gel was transferred to a PVDF membrane (Labiskoma, Seoul, Republic of Korea, KDM50) at 100 V for 1 h. The membrane was blocked with 5% nonfat milk (Bio-Rad, BR1706404) in TBST for 1 h with gentle shaking. The membrane was incubated overnight at 4 °C with LC3A/B (D3U4C) (Cell Signaling Technology, Danvers, MA, USA, 12741) in a blocking buffer. After incubation, the membrane was washed three times in 1× TBST for 10 min. It was incubated with Goat anti-rabbit IgG (ENZO, ADI-SAB-300-J) for 1 h and washed three times in 1× TBST for 10 min. The images were obtained by Davinch-Chemi Fluro imager (Davinch-K, Seoul, Republic of Korea) after incubation with SuperSignal™ West Pico PLUS Chemiluminescent Substrate (Thermo Scientific™, 34578). 

### 2.9. Visualization of LC3 Expression on NPs

LC3 expression was visualized by confocal microscopy and live cell video. In confocal microscopy, the 1 × 10^6^ LC3 cells were seeded on a microscope cover glass (SUPERIOR, Soothfield, MI, USA, HSU-0101050) in a 6-well plate and stabilized for 24 h. The old media were discarded along with 50 µg/mL Fe_3_O_4_, SiO_2_, and these mixtures in the media were added to the well. After 24 h of incubation, the NPs were carefully removed and washed with PBS. Then, 1 mL of 4% Paraformaldehyde (BIOSOLUTION, Seoul, Republic of Korea, BP031) was added and washed with cold PBS after incubation for 15 min. DAPI (Sigma-Aldrich, D9542) was diluted 10,000 times and incubated for 15 min in the dark. The slides were mounted with Fluoromount-G™ Mounting Medium (Invitrogen™, 00-4958-02). The images were obtained with FLUOVIEW FV3000 (Olympus, Tokyo, Japan).

Live cell videos were taken using Celloger Mini Plus (Curiosis, Seoul, Republic of Korea). The LC3 cells were seeded at 1 × 10^6^ cells/well on a 6-well plate and stabilized for 24 h. The media were changed carefully to new media containing 50 µg/mL Fe_3_O_4_, SiO_2_, and these former mixtures. The videos were recorded every 10 min for 24 h. 

### 2.10. Analysis of Data

The data were analyzed and graphed using GraphPad Prism software version 9.4 (GraphPad Software Inc., San Diego, CA, USA). In addition, *t*-test and one-way ANOVA were used to evaluate the *p*-value and the value was marked as less than 0.05. All relative values were normalized with negative control, which did not treat nanoparticles. 

## 3. Results and Discussions

The materials used in this research were SiO_2_ with a size of 20 (20 ± 5) nm in diameter, and Fe_3_O_4_ with a size of around 20 (20 ± 5) nm. The materials were synthesized and characterized using traditional chemical and physical methods [49]. Physical characterization and electron microscopy graphs can be seen in Figure 1 and EDAX analysis of SiO_2_ presented in Appendix A. This synthesis method and its detailed properties were reported in earlier publications [50,51,52]. The mesoporous SiO_2_ nanoballs and Fe_3_O_4_ were of uniform size and shape, an appropriate material for the next step in the experiments.

For assessments of autophagy modulation, the LC3 transfected cells were generated. The transfection was successfully confirmed in RT–qPCR, increasing LC3 gene levels only in LC3 transfected cells (Figure 2A). The cytotoxicity studies of Fe_3_O_4_, SiO_2_ and these mixtures were characterized using WST-8 assay with LC3 transfected cells. Earlier results confirmed that kidney cells were competent in up-taking NPs through the membrane, which strengthened the NP concentrations of the intracellular compartments. Aging-related inflammatory stresses of the kidney were protected by autophagy [53]. Under autophagy–deficient conditions, degeneration of the kidney occurred [54,55,56]. Factors in apoptosis and fibrosis increased the production of ROS from kidney injury and chronic inflammation to kidney senescence [57]. A series of particles at concentrations from 1 to 100 μg/mL was measured cytotoxic influences. The Fe_3_O_4_, SiO_2_, and these mixtures did not show any toxicity in various concentrations (Figure 2B). The ROS levels of Fe_3_O_4_, SiO_2_, and these mixtures were similar to the negative control, and significantly lower than the positive control (H_2_O_2_ 400 µM; Figure 2C). 

Western blot and RT–qPCR were performed to confirm the modulation of autophagy by analyzing the change in LC3. In Figure 3A, the LC3-II appeared as a dark band in Fe_3_O_4_, SiO_2_, and their mixture. The total LC3 and LC3-II levels increased after treatment of Fe_3_O_4_, SiO_2_, and their mixture. Interestingly, the LC3-II/Ⅰ level, indicating LC3 activation, also increased following nanoparticle treatment (Figure 3B). The increased intensity of LC3 post-nanoparticle treatment indicates potential autophagy enhancement. Additionally, the levels of LC3 gene were elevated, except for Fe_3_O_4_ (Figure 3C). Although there was no correlation between the western blot and RT-qPCR results, an increase in LC3 was observed for both SiO_2_ and the Fe_3_O_4_ + SiO_2_ mixture. This suggests the possibility of alternative pathways influencing LC3 dynamics beyond canonical autophagy [58].

The confirmation of LC3 changes was performed via confocal and live cell monitoring. The intense green fluorescence observed after nanoparticle treatment indicates strong LC3 expression (Figure 4, S2SI1), confirming autophagosome formation. The green fluorescence is notably more intense in SiO_2_ and the Fe_3_O_4_ and SiO_2_ mixture. The green fluorescence in live cell monitoring increased in intensity from 0 h, surpassing that of the negative control.

We represent the productive autophagosome mechanism in Figure 5. This result proved that the SiO_2_ and Fe_3_O_4_ NPs influenced the autophagosome formation. The addition of SiO_2_ and Fe_3_O_4_ could generate the autophagy process inside the cells. The autophagic potential was activated in response to stress-induced activity by SiO_2_ and Fe_3_O_4_ nanoparticles. The continuous activation of autophagy facilitated cell proliferation and contributed to kidney repair. Overall, this finding proved a novel pathway for autophagy modulation, which was not known previously.

## 4. Conclusions 

Since autophagy has been proven to be involved in chemotherapy and radiotherapy, researchers could target the modulation of autophagy by NPs as an attractive new therapy. Researchers investigated Fe_3_O_4_ NPs and SiO_2_ NPs to determine if they might improve kidney cell proliferation and repair kidney injury. The current result illustrated that Fe_3_O_4_ and SiO_2_ NPs could promote cell growth and modulate autophagy. Especially, the mixture of Fe_3_O_4_ and SiO_2_ NPs induces the autophagy process for stimulation of and enhancement of cell growth. We will further investigate the molecular links/pathways between Fe_3_O_4_ and SiO_2_ for their synergistic activity to understand their mechanisms. Indeed, modulating autophagy could help to enhance its therapeutic effects.

## Figures and Tables

**Figure 1 nanomaterials-14-01033-f001:**
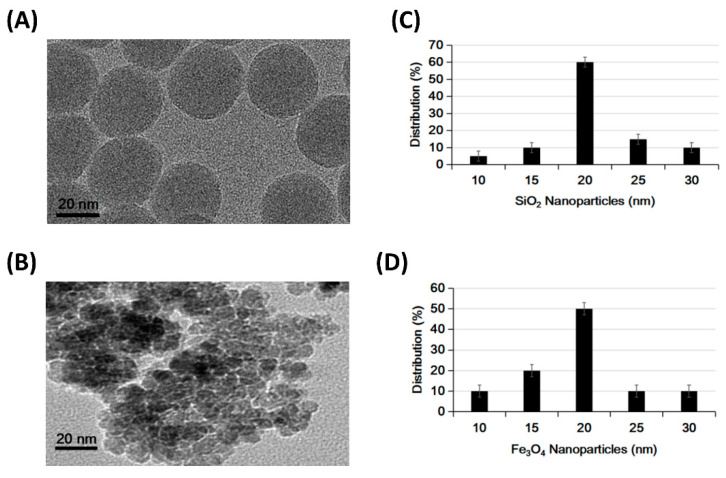
TEM images of both (**A**) SiO_2_ nanoparticle and (**B**) Fe_3_O_4_ nanoparticle, respectively. The corresponding size distribution histogram for both (**C**) SiO_2_ nanoparticle and (**D**) Fe_3_O_4_ nanoparticle, respectively. The scale bar is 20 nm.

**Figure 2 nanomaterials-14-01033-f002:**
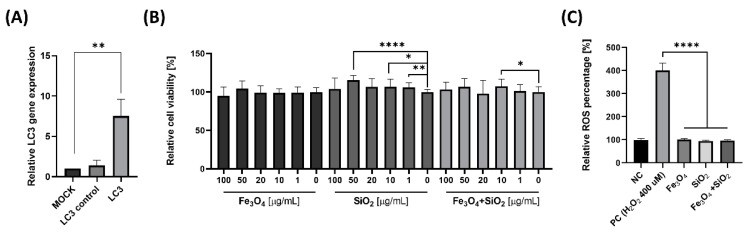
LC3 transfected cells were used for the cell experiments. (**A**) Levels of the LC3 gene. (**B**) Testing the cytotoxicity of different concentrations (0–100 µg/mL) of Fe_3_O_4_, SiO_2_, and their combination. (**C**) ROS levels at 50 µg/mL for Fe_3_O_4_, SiO_2_, and their mixture. The data are reported as mean ± SD (*n* = 3). The data were standardized using a negative control that did not involve nanoparticle treatment. * *p* < 0.05, ** *p* < 0.005, **** *p* < 0.0001.

**Figure 3 nanomaterials-14-01033-f003:**
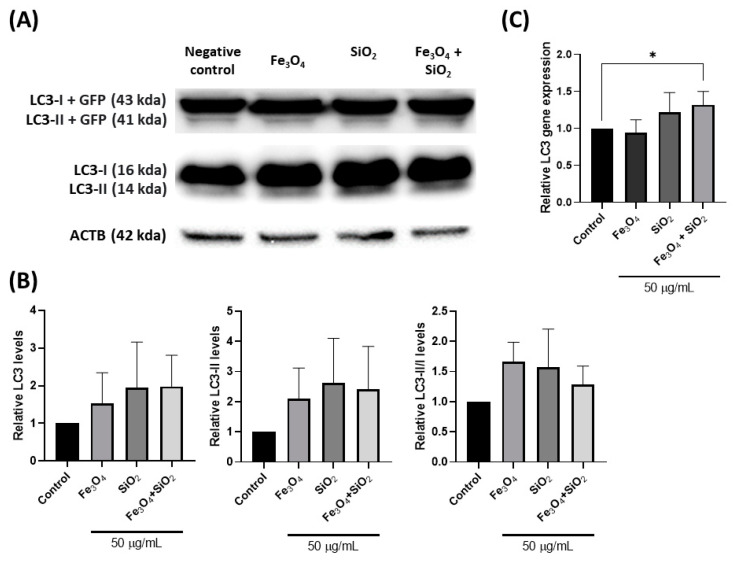
Alteration of LC3 treatment using nanoparticles. (**A**) Western blot images of nanoparticles treated at 50 µg/mL for 24 h. (**B**) Relative band intensity of the western blot. (**C**) Expression levels of LC3 gene at 50 µg/mL for Fe_3_O_4_, SiO_2_, and their combination. All data are presented as mean ± SD (*n* = 3). The data were normalized using a negative control that did not involve nanoparticle treatment. * *p* < 0.05.

**Figure 4 nanomaterials-14-01033-f004:**
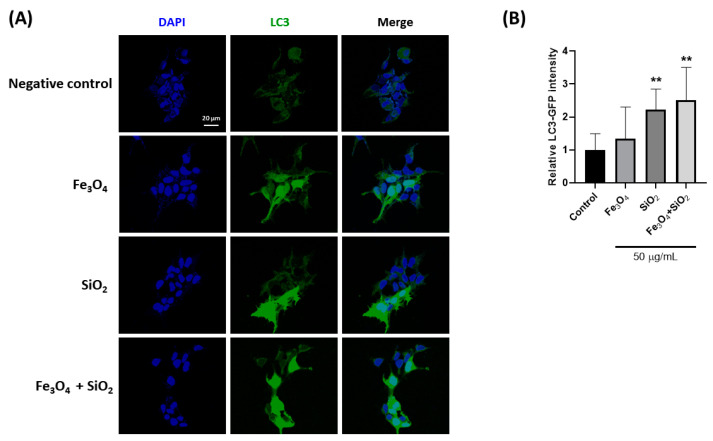
Fluorescence analysis of LC3 marker (green) and nucleus (blue, DAPI) in Fe_3_O_4_, SiO_2_, and their combination nanoparticles. (**A**) Fluorescence images (**B**) The intensity of the fluorescence signal Scale bar: 20 µm. The intensity was calculated dividing green into blue, and data were normalized using a negative control that did not involve nanoparticle treatment. ** *p* < 0.005.

**Figure 5 nanomaterials-14-01033-f005:**
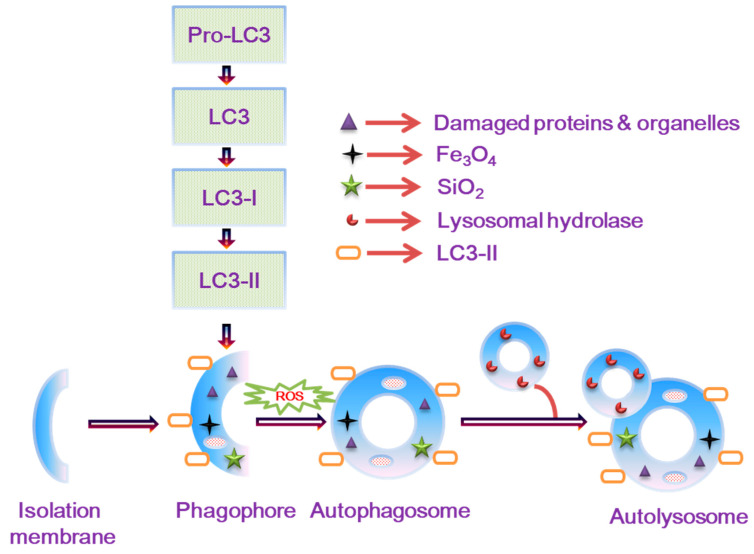
Schematic description of the autophagy mechanism induced by Fe_3_O_4_ and SiO_2_ NPs.

## Data Availability

The original contributions presented in the study are included in the article/Appendix A, further inquiries can be directed to the corresponding author.

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
