# Peer review of "Synergistic Effect of SiO2 and Fe3O4 Nanoparticles in Autophagy Modulation"

_nanomaterials, 2024, doi:10.3390/nano14121033_

Round 1

Reviewer 1 Report

Comments and Suggestions for Authors

Comments on the Quality of English Language

The English editing of the overall manuscript should be improved before publication.

Author Response

Reviewer 1

This manuscript investigates the “Synergistic Effect of SiO2 and Fe3O4 nanoparticles in autophagy modulation”. The overall article lacks some logical discussions and after the revision, the manuscript can be accepted for publication in Nanomaterials.

  1. The author should draw at least one schematic to summarize the overall conclusion of the paper for a better understanding of the readers.

Answer: Thank you for your review. We already have schematic figure (Figure 5) in our paper and could think it is helpful to understand our overall discussion.

  1. The introduction should clearly state the novelty and purpose of this review article.

Answer: Thank you for your comment. We have revised the manuscript.

  1. What could be th main reasons for SiO2 and Fe3O4 influential effect on autophagosome formation as shown in Fig. 5?

Answer: Thank you for your review for Figure 5. We thought that the primary reasons for the observed effects are the following: The high surface area-to-volume ratios of SiO2 and Fe3O4 nanoparticles could increase their reactivity and induce oxidative stress, which can trigger autophagy. Additionally, the size and morphology of these nanoparticles influence cellular uptake and intracellular trafficking, affecting autophagosome formation. Fe3O4 nanoparticles catalyze ROS formation via Fenton reactions, while SiO2 nanoparticles induce ROS production through different mechanisms, and both types of ROS generation can activate autophagy. Nanoparticles are internalized through endocytosis and accumulate in lysosomes, disrupting their function and promoting autophagosome formation.

  1. Can we distinguish a specific role of Fe3O4 by comparing the results of SiO2 and Fe3O4+SiO2 in Fig. 3b?

Answer: Thank you for your comment. In these results, we aimed to demonstrate that nanoparticle treatment leads to sustained LC3 activity, as evidenced by the increase in gene expression and supported by Western blot results. Our findings show a correlation between western blot (Fig 3a) and gene expression level (Fig 3b). When comparing treatments, we observed that the combined Fe3O4 and SiO2 treatment resulted in higher LC3 expression than either Fe3O4 or SiO2 alone. Furthermore, we can distinguish a specific role of Fe3O4 by comparing the results of SiO2 and Fe3O4+SiO2 in Fig. 3b. The combined treatment (Fe3O4+SiO2) led to a significantly higher increase in LC3 gene expression compared to SiO2 alone, indicating that Fe3O4 has a synergistic effect when mixed with SiO2. This suggests that Fe3O4 can enhance the autophagic response, but treatment with Fe3O4 together with SiO2 may increase the autophagic response more than treatment alone.

  1. Fig. 1b is not clear to make any significant conclusion. The author should add more images in the supporting information.

Answer: Thank you for your comment. We have added its analysis in supporting information.

  1. It is better to revise the conclusion based on future perspectives and recommendations and the research gap between the present and commercial scale applications.

Answer: Thank you for your precious comment. We have revised the manuscript.

  1. I suggest authors add a comparison table based on the comparison of previous literature.

Answer: Thank you for the review. We have revised the manuscript.

After revision:

Maria et. al. [45] demonstrated that dextran-coated iron oxide nanoparticles have anti-inflammatory activity at non-cytotoxic concentrations by reducing the levels of pro-inflammatory mediators such as IL-1β, MCP-1, CCR2, TNF-α, and IL-6 in activated endothelial cells and M1 phenotype macrophages. Folic acid-coated iron oxide nanoparticles enhance internalization and facilitate delivery of therapeutic agents for cancer treatment and inflammation-related diseases like rheumatoid arthritis, lupus, osteoarthritis, Crohn's disease, and atherosclerosis [46]. 

  1. Is there any effect of SiO2 an Fe3O4 particle size over the kidney cell proliferation as mentioned in the conclusion.

Answer: Thanks for your insight feedback. We thought that the size of nanoparticles could influence their biological interactions and effects. Smaller particles generally have a higher surface area-to-volume ratio, which can enhance their reactivity and interaction with cellular components, potentially leading to greater effects on cell proliferation. Larger particles, on the other hand, might have different uptake dynamics and cellular responses. In the context of our study, we focused on the autophagy-related effects of SiO2 and Fe3O4 nanoparticles. So, in the future, it would be beneficial to conduct additional experiments specifically investigating how different sizes of these nanoparticles affect kidney cell proliferation to comprehensively address this aspect.

  1. The English editing of the overall manuscript should be improved before publication. In addition, 23% plagiarism was detected in the overall manuscript which should be reduced to less than 10%, to maintain the journal's reputation and be considered as a high impact in the literature.

Answer: Thank you for your detailed review. We have revised address some of the issues raised. Regarding the detected plagiarism, we acknowledge the concern and have taken steps to reduce it. We understand that the similarity in methodology to our previous studies may have contributed to the detected plagiarism. However, we have carefully revised our manuscript and incorporated additional writing based on the insightful feedback provided by the reviewers.

Reviewer 2 Report

Comments and Suggestions for Authors

The manuscript entitled Synergistic Effect of SiO2 and Fe3O4 nanoparticles in autophagy modulation by Sitansu Sekhar Nanda et al is well presented and aims to understand the changes of induction and perturbation of autophagy in HEK-293 kidney cells by iron oxide (Fe3O4) and silica ( SiO2) nanoparticles. This topic is of current interest and the problem the authors pose is real and urgent.

The introduction section is well written.

However, the authors can cite the new works that use Fe3O4 magnetic nanoparticles published recently.

1.       Pharmaceutics. 2021 Sep 7;13(9):1414. doi: 10.3390/pharmaceutics13091414.

2.       Int J Pharm. 2022 Sep 25;625:122064. doi: 10.1016/j.ijpharm.2022.122064.

The materials and methods section must present the independent experiments performed for each test. If the authors use only n=3 data for average calculation, it is not a proper way to show a cellular response and sustain the manuscript's conclusion. It is necessary to do at least 3 independent experiments each in triplicate.

The results section is well presented, still, there are some recommendations:

-          please mention the number of independent experiments in each Figure description.

-          Figure 3 - The protein expression for LC3-I/LC3-II-GFP and total form was only demonstrative. The western blots must be accompanied by the average data obtained by densitometry and the corresponding histogram to compare with the molecule's gene expression and sustain the manuscript conclusion.

Author Response

Reviewer 2

The manuscript entitled Synergistic Effect of SiO2 and Fe3O4 nanoparticles in autophagy modulation by Sitansu Sekhar Nanda et al is well presented and aims to understand the changes of induction and perturbation of autophagy in HEK-293 kidney cells by iron oxide (Fe3O4) and silica ( SiO2) nanoparticles. This topic is of current interest and the problem the authors pose is real and urgent.

The introduction section is well written.

However, the authors can cite the new works that use Fe3O4 magnetic nanoparticles published recently.

  1. Pharmaceutics. 2021 Sep 7;13(9):1414. doi: 10.3390/pharmaceutics13091414.
  2. Int J Pharm. 2022 Sep 25;625:122064. doi: 10.1016/j.ijpharm.2022.122064.

Answer: Thank you for your consideration. We added your recommendation in ~

After revision:

Maria et. al. [45] demonstrated that dextran-coated iron oxide nanoparticles have anti-inflammatory activity at non-cytotoxic concentrations by reducing the levels of pro-inflammatory mediators such as IL-1β, MCP-1, CCR2, TNF-α, and IL-6 in activated endothelial cells and M1 phenotype macrophages. Folic acid-coated iron oxide nanoparticles enhance internalization and facilitate delivery of therapeutic agents for cancer treatment and inflammation-related diseases like rheumatoid arthritis, lupus, osteoarthritis, Crohn's disease, and atherosclerosis [46]. 

The materials and methods section must present the independent experiments performed for each test. If the authors use only n=3 data for average calculation, it is not a proper way to show a cellular response and sustain the manuscript's conclusion. It is necessary to do at least 3 independent experiments each in triplicate.

The results section is well presented, still, there are some recommendations:

-          please mention the number of independent experiments in each Figure description.

Answer: Thank you for your feedback. Our data of this paper were analyzed after independence experiments in triplicate. We calculated average based on triplicate of independent experiments.

-          Figure 3 - The protein expression for LC3-I/LC3-II-GFP and total form was only demonstrative. The western blots must be accompanied by the average data obtained by densitometry and the corresponding histogram to compare with the molecule's gene expression and sustain the manuscript conclusion.

Answer: Thanks for your reviews. We added the quantification of western blot. Furthermore, as your feedback, we mentioned correlation between western blot and gene expression in the result part.

Reviewer 3 Report

Comments and Suggestions for Authors

This manuscript explores the influence of SiO2 and Fe3O4 on the autophagy of HEK-293 kidney cells. The manuscript would benefit from addressing the following points to provide a clearer and more detailed presentation of the study's findings and implications.

P2 Line 48: There's an extra space between "of" and "SiO2."

The authors should provide background information or a rationale for choosing HEK-293 kidney cells for this study.

Ensure there is a space between the number and the unit throughout the manuscript.

Figure 1: Clarify how the size distribution histograms were created. If the scale is correct, the TEM images show that SiO2 and Fe3O4 particles are obviously different in size.

Figure 4: Is there a way to quantify the green fluorescence? The SiO2 and Fe3O4/SiO2 samples appear to have the same green fluorescence intensity.

Figure 5: While Figure 5 aims to describe the autophagy mechanism induced by the particles, more detailed information should be included in the paper.

The conclusion states a synergistic effect of SiO2 and Fe3O4. However, the manuscript does not provide clear evidence that SiO2/Fe3O4 is superior to Fe3O4 or SiO2 alone.

Comments on the Quality of English Language

Check the typos throughout the paper.

Author Response

Reviewer 3

This manuscript explores the influence of SiO2 and Fe3O4 on the autophagy of HEK-293 kidney cells. The manuscript would benefit from addressing the following points to provide a clearer and more detailed presentation of the study's findings and implications.

P2 Line 48: There's an extra space between "of" and "SiO2."

The authors should provide background information or a rationale for choosing HEK-293 kidney cells for this study.

Answer: Thank you for your feedback. We chose to use HEK293 cells for LC3 transfection and to evaluate the enhancement of autophagy function by nanoparticles for following reasons. The HEK-293 cells were easy to transfection and broad utilization in many studies (Phadwal K et al., 2012; Jiang H, et al., 2010; Sofia V. Salvatore et al., 2024). Furthermore, if our focus had been on neurodegenerative diseases, we would have used a cell line such as SH-SY5Y. However, our current study aims to examine the changes in autophagy function induced by nanoparticles. Therefore, we chose to use HEK-293 cells, which are widely used and well-suited for general studies.

Ensure there is a space between the number and the unit throughout the manuscript.

Answer: Thank you for your detail consideration. We checked it once again.

Figure 1: Clarify how the size distribution histograms were created. If the scale is correct, the TEM images show that SiO2 and Fe3O4 particles are obviously different in size.

Answer: Thanks for your comments. We have revised the manuscript. We have added supporting information.

Figure 4: Is there a way to quantify the green fluorescence? The SiO2 and Fe3O4/SiO2 samples appear to have the same green fluorescence intensity.

Answer: Thank you for your feedback. We calculated the florescence. We normalized the green florescence to divide the DAPI florescence signal.

Figure 5: While Figure 5 aims to describe the autophagy mechanism induced by the particles, more detailed information should be included in the paper.

The conclusion states a synergistic effect of SiO2 and Fe3O4. However, the manuscript does not provide clear evidence that SiO2/Fe3O4 is superior to Fe3O4 or SiO2 alone.

Answer: We appreciated your precious comment. Adding the quantifying data from western blot and florescence intensity could enhance our discussion. Because the quantifying data showed strong intensity after treating mix of Fe3O4 and SiO2.

Reviewer 4 Report

Comments and Suggestions for Authors

1. Review of the bibliography with new scientific works (2021-2024) to see if there is progress compared to 2020.

2. Since the quality and purity of Fe3O4 and SiO2 nanoparticles is crucial for such uses, I request their EDAX and RDX spectra. To interpret and discuss the results.

3. The size of the nanoparticles is extremely important for these applications, I request the specification of the size of the nanoparticles from the RDX spectra. Specify whether they are appropriate for the purpose of the work.

Author Response

Reviewer 4

  1. Review of the bibliography with new scientific works (2021-2024) to see if there is progress compared to 2020.

Answer: Thank you for your feedback. We have mentioned in introduction section.

After revision:

Maria et. al. [45] demonstrated that dextran-coated iron oxide nanoparticles have anti-inflammatory activity at non-cytotoxic concentrations by reducing the levels of pro-inflammatory mediators such as IL-1β, MCP-1, CCR2, TNF-α, and IL-6 in activated endothelial cells and M1 phenotype macrophages. Folic acid-coated iron oxide nanoparticles enhance internalization and facilitate delivery of therapeutic agents for cancer treatment and inflammation-related diseases like rheumatoid arthritis, lupus, osteoarthritis, Crohn's disease, and atherosclerosis [46]. 

  1. Since the quality and purity of Fe3O4 and SiO2 nanoparticles is crucial for such uses, I request their EDAX and RDX spectra. To interpret and discuss the results.

Answer: Thanks for your recommendation. We have mentioned in supporting information.

  1. The size of the nanoparticles is extremely important for these applications, I request the specification of the size of the nanoparticles from the RDX spectra. Specify whether they are appropriate for the purpose of the work.

Answer: Thank you for your comment in our study. We have checked its size in zeta sizer.

Round 2

Reviewer 3 Report

Comments and Suggestions for Authors

The authors have addressed most of my comments. I could not find Figure S1 of SiO2 EDAX. I am not convinced that SiO2 and Fe3O4 are comparable in size.

Author Response

Dear Reviewer,

Thank you for consideration. Please go through the attached file.

Thanking You

Reviewer 4 Report

Comments and Suggestions for Authors

Affirm the following:

... and EDAX analysis of SiO2 presented in figure S1...

Please help us identify figure S1.

Author Response

(The authors gave the same response as above.)
